# Association between Intracellular Calcium Signaling and Tumor Recurrence in Human Non-Functioning Pituitary Adenomas

**DOI:** 10.3390/ijms25073968

**Published:** 2024-04-03

**Authors:** Yorgui Santiago-Andres, Ana Aquiles, Keiko Taniguchi-Ponciano, Latife Salame, Gerardo Guinto, Moises Mercado, Tatiana Fiordelisio

**Affiliations:** 1Laboratorio de Neuroendocrinología Comparada, Facultad de Ciencias, Universidad Nacional Autónoma de México, Ciudad Universitaria, Ciudad de Mexico 04510, Mexico; sayorgui@ciencias.unam.mx; 2Posgrado en Ciencias Biológicas, Unidad de Posgrado, Edificio D, 1° Piso, Circuito de Posgrados, Ciudad Universitaria, Ciudad de Mexico 04510, Mexico; 3Instituto de Neurobiología, Universidad Nacional Autónoma de México, Campus Juriquilla, Querétaro 76230, Mexico; anaaquiles@ciencias.unam.mx; 4Unidad de Investigación Médica en Enfermedades Endocrinas, Hospital de Especialidades, Centro Médico Nacional Siglo XXI, Instituto Mexicano del Seguro Social, Ciudad de Mexico 06720, Mexico; keiko.taniguchi@hotmail.com (K.T.-P.); salamekhouri.latif@gmail.com (L.S.); 5Centro Neurológico, Centro Médico ABC, Ciudad de México 05370, Mexico; gguinto@prodigy.net.mx

**Keywords:** PitNET, CNFPA, hormone secretion, calcium, cell signaling, endocrine disruption, non-secreting pituitary adenoma, silent pituitary adenoma

## Abstract

Clinically non-functioning pituitary adenomas (CNFPAs) are the second most frequent sellar tumor among studies on community-dwelling adults. They are characterized by the absence of hormonal hypersecretion syndrome, and patients present with compressive symptoms, such as a headache and visual field defects. Immunohistochemically, most CNFPAs are of gonadotrope differentiation, with only a few of them being truly null cell adenomas. Although these tumors express receptors for one or more hypothalamic releasing hormones, to what extent this has an impact on the biological and clinical behavior of these neoplasms remains to be defined. In this research, we evaluated the basal and hypothalamic secretagogue-stimulated intracellular calcium mobilization in 13 CNFPAs, trying to correlate this response to the phenotypic features of the patients. Our results indicate that the recurrence of a CNFPA correlates positively with cellular responsiveness, as measured by spontaneous intracellular calcium activity and the ability to respond to multiple hypothalamic secretagogues. We conclude that this finding may be a useful tool for predicting the clinicopathologic behavior of CNFPAs, by testing the variation of cellular responsiveness to hypothalamic secretagogues.

## 1. Introduction

Pituitary adenomas constitute 15% of all intracranial neoplasms [1]. Among studies on community-dwelling adults, clinically non-functioning pituitary adenomas (CNFPAs) represent 33% of all intrasellar lesions and are the most common type of adenoma when considering only macroadenomas (i.e., tumors larger than 1 cm) [2,3]. Even though over 60% of CNFPAs are of gonadotrope differentiation, they seldom result in hormonal hypersecretion syndrome [2,3]. Approximately 20% of CNFPAs are silent adenomas that immunostain for ACTH, GH, PRL, and TSH, or a variable combination of these hormones. Less than 20% of CNFPAs are real, null cell adenomas that do not immunostain for any hormone, or express pituitary-specific transcription factors [2,3]. The diagnosis of CNFPAs is either made incidentally or relies on the detection of compressive symptoms and signs, such as headaches and visual field abnormalities due to compromise of the optic chiasm [2].

Many years ago, it was established that hormonal secretion of the different pituitary cell populations (gonadotrophs, somatotrophs, lactotrophs, corticotrophs, and thyrotrophs) is dynamically regulated by one of the neurohormones produced in the hypothalamus (GnRH, GHRH, DA, CRH, and TRH, respectively) [4,5]. Traditionally, it was thought that each of these highly differentiated cells specifically respond to the pulsatile secretion of their specific hypothalamic-releasing hormone. We now know that the pituitary gland has evolved into an organ of high plasticity in order to adapt to different physiological, developmental, and pathological conditions [6]. These cellular plasticity processes include the modification of cellular responsiveness to canonical and non-canonical secretagogues, the synchronization of cell signaling, transdifferentiation, and communication between intertwined homotypic and heterotypic cellular networks [7,8,9,10]. Dysfunction of these processes has implications for endocrine homeostasis and diseases that could include CNFPAs. CNFPAs may present an imbalance in these cellular plasticity processes, since it has been observed that in biopsy-derived primary cell cultures, there is an apparent paradoxical secretion, involving the response of a particular cell type to certain non-canonical hypothalamic stimulus [11,12,13]. Cells are able to respond to more than one secretagogue (multi-responsive cells) and other cells synthesize more than one hormone (multi-hormonality) [11]. Furthermore, even though these observations have been reported to be common in CNFPAs, patients frequently develop pituitary hormone deficiencies (such as TSH) [14,15].

The Intracellular signaling responsible for hormone secretion is quite similar among the different neuroendocrine systems [4,16,17,18,19]. This process involves the activation of G proteins through specific cell membrane receptors by hypothalamic-releasing hormones. Subsequently, the activation of phospholipase C (PLC) promotes the production of inositol-3-phosphate (IP_3_) and diacylglycerol, as well as the concomitant activation of protein kinase C (PKC) to finally increase the intracellular calcium concentration ([Ca^2+^]_i_), which allows vesicular transport and fusion to the membrane and, ultimately, hormone secretion to the hypophyseal portal system. Although modification to intracellular calcium signaling in tumor cells could provide clues to the mismatch between stimulation and secretion, this process is not well understood and needs further study [20,21]. In this work, we investigated intracellular Ca^2+^ mobilization (Ca^2+^_i_) in CNFPAs to elucidate a possible hallmark of common features among them, as well as to predict the clinicopathological course that a tumor may follow, according to its cellular responsiveness.

## 2. Results

### 2.1. Patients’ Characteristics

The CNFPAs of 13 subjects were included in this study (Table 1). Their mean age was 53.6 ± 9.5 years, and eight of them were women. All had compressive symptoms, such as a headache and visual field abnormalities, but no evidence of hormonal hypersecretion syndrome. All but one subject (tumor 8) had at least one pituitary hormone deficiency at diagnosis; central hypothyroidism was documented in 12 patients (92.3%), central hypocortisolism in three patients (23%), and central hypogonadism in four patients (30.7%). A combined TSH and ACTH deficiency was present in three patients (23%), whereas panhypopituitarism involving all pituitary hormone systems was documented in five patients (38.4%).

All subjects harbored pituitary macroadenomas with a mean maximal tumor diameter of 4.7 ± 1.49 cm. Cavernous sinus invasion was evident in the magnetic resonance imaging (MRI) of nine subjects (69.2%). Transsphenoidal surgery (TSS) was the primary treatment in eleven subjects (84.6%), while three (23%) required a transcranial (TC) approach, and five had tumor recurrence (38.5%) requiring two or more surgical procedures, with subject 2 having up to six surgical procedures. Postoperative radiation therapy was administered to three subjects (23%), three subjects were treated with cabergoline, and one received temozolomide. For recurrent CNFPAs, we evaluated the last event of recurrence in this work, as indicated in Table 1.

Ten adenomas (77%, tumors 2, 4–12) were classified as gonadotrophinomas, based on positive immunostaining for the alpha subunit of the glycoprotein hormones, LHβ and/or FSHβ, as well as for the lineage-determining transcription factor NR5A1 (also known as SF-1); two tumors (15.4%, tumors 1 and 3) were categorized as null cell adenomas, since they did not immunostain for any pituitary hormone or transcription factor; and one (7.7%, tumor 13) of the tumors was considered to be a silent corticotroph adenoma because it immunostained for ACTH and the lineage-specific transcription factor TBX19 (also known as T-Pit). Table 1 summarizes these immunohistochemical findings.

### 2.2. Vasculature

The vascular architecture varied considerably among the tumors. Some tumors displayed a low vascular density (tumor 6 and 12, Figure 1), which correlated with a reduced number of capillary branching events. Tumors 2 and 9, on the other hand, were highly vascularized, with rich branching and vessel density. Interestingly, these patterns of vessel architecture determine the cellular distribution within the tissue. The honeycomb-like arrangement normally found in non-tumoral pituitary tissue was present only in tumors 2 and 9 (Figure 1), while the remaining majority displayed a highly disorganized vasculature, with a central core of vessels and long, thin capillaries, with most cells located far away from them (Figure 1).

### 2.3. Basal Intracellular Calcium Activity

To assess cellular changes in [Ca^2+^]_i_ tumor tissues were challenged by hypothalamic hormones. Cells that responded to membrane depolarization with a high potassium solution and mobilized Ca^2+^ were considered alive and represented 100% of the analyzed cells and were reported per tumor. As summarized in Figure 2, a portion of the tumor cells exhibited small spontaneous increases in [Ca^2+^]_i_ without any external stimulation.

The adenomas were first classified by their spontaneous Ca^2+^ activity (Figure 2A–G and Appendix A). Tumors 4, 7, 8, 9, and 11 showed no spontaneous [Ca^2+^]_i_ activity. In tumors 1, 3, 5, 6, and 10, spontaneous [Ca^2+^]_i_ activity was observed in less than 20% of the cells, comprising 15.6%, 2.8%, 11.7%, 5.2%, and 6.4%, respectively; whereas, in tumors 2, 12, and 13, more than 20% of the cells had basal activity (57.4%, 34.8%, and 32.3%, respectively).

While the spontaneous Ca^2+^ activity in non-recurrent tumors was null or very limited (less than 20% of cells), more than 20% of cells in the recurrent tumors were capable of mobilizing intracellular Ca^2+^ (Figure 3A). The tumor with the highest spontaneous intracellular Ca^2+^ activity, tumor 2, was the most recurrent in the series, since the patient required six transsphenoidal surgeries, adjunctive radiation therapy, as well as pharmacological therapy with cabergoline and even temozolomide for growth control. Conversely, tumor 8, without spontaneous intracellular Ca^2+^, is a small non-recurrent tumor (2.6 cm) with no evidence of hormonal deficiency.

Mobilization of spontaneous [Ca^2+^]_i_ activity is heterogeneous among the cells, with some cells presenting a single but prolonged Ca^2+^ mobilization event during recording (Figure 2F), while others presented rapidly increasing and decreasing peaks, varying between one and up to 12 events (Figure 2C–E). Cells with one or two calcium events were the most frequent. Tumors 1 and 3 had the highest mean frequency of [Ca^2+^]_i_ mobilization events, 2.3 Hz and 2.59 Hz, respectively, and expressed all the progenitor markers evaluated (*NR5A1*, *POU1F1*, and *TBX19*). Additionally, the tumors from all cabergoline-treated patients also showed spontaneous [Ca^2+^]_i_ calcium activity.

### 2.4. Secretagogue-Induced Intracellular Calcium Activity

Even though all adenomas were classified as CNFPAs, they show a wide variation in regard to the Ca^2+^_i_ activity and secretagogue response (Figure 2H–O). The tumor cells were categorized as non-responsive (neither spontaneous activity, nor a response to secretagogues, but their viability was confirmed by stimulation with a high potassium solution; Figure 2I), mono-responsive (cells responding to a single secretagogue; Figure 2J), or multi-responsive (cells responding to more than one secretagogue; Figure 2K–N), according to [Ca^2+^]_i_ mobilization elicited by exposure to different hypothalamic secretagogues. The proportion of tumors in each category was analyzed and compared (Figure 2O).

We found that non-recurrent CNFPAs are also characterized by a predominant percentage of non-responsive cells (more than 50%, see Figure 3B and Appendix A) to any hypothalamic secretagogue and, therefore, were classified as overwhelmingly non-responsive tumors, this category being represented by tumors 4 (97.9%), 7 (97.8%), 11 (86.9%), 3 (76%), 9 (75%), 10 (61.7%), and 1 (56%). On the other hand, recurrent CNFPA 2, 5, 6, 12, and 13 have less than 50% of non-responsive cells and a significant proportion of mono- and multi-responsive cells, so were classified as overwhelmingly responsive (Figure 3C,D). The only null cell adenoma (tumor 7) was completely non-responsive (Figure 2O).

Interestingly, mono- and multi-responsive tumors were not necessarily more responsive to the putatively canonical secretagogue, according to their classification in Table 1 (see also Figure 2O). Among the eight sensitive gonadotropinomas known to express the GnRH receptor, only cells from three tumors (6, 10, 12) showed a predominant response to GnRH (Figure 2O), whereas the remaining five responded preferentially to other secretagogues, such as GHRH, TRH, and CRH (Figure 2O). Of the three “silent” adenomas, tumor 1, a silent GH adenoma, responded predominantly to GHRH and to a lesser extent to GnRH and TRH, whereas tumor 13, a silent ACTH adenoma, responded similarly to GHRH, CRH, TRH, and TRH/DA. Tumor 8, a silent plurihormonal adenoma, responded exclusively and intensely to TRH/DA, suggesting a predominant lactotroph phenotype-like differentiation (Figure 2O).

The tumor responsiveness to hypothalamic hormones varied between subjects. Tumor 8 was the smallest non-recurrent silent CNFPA and, although most of its cells were responsive (Figure 2O), they mostly responded to a single secretagogue; the percentage of non- and multi-responsive cells was low (3.16 and 0.22%, respectively) and the predominant response was mono-responsive (96.3%). In contrast, tumor 2, a highly recurrent CNFPA (six surgeries) had a predominantly multi-responsive activity (57.1%) to hypothalamic hormones and a very low mono-response (15.9%). In addition, an intermediate state was represented by tumor 6, which was a recurrent tumor with a small proportion of spontaneous intracellular Ca^2+^ activity and less than 20% of non-responsive cells, and tumor 8, a silent GH/PRL/LH/FSH/TSH/ACTH adenoma with a proportion of mono responsive cells higher than 50% (Figure 2O and Figure 3).

Based on the heterogeneity of cell responsiveness across the analyzed CNFPAs, we hypothesized that this variation could represent the potential of cell multi-responsiveness and may be associated with clinical data. We performed a principal component analysis to project the data on spontaneous calcium activity, the response to hypothalamic secretagogues, and the adenoma size (Appendix A) into a set of principal factors that explain the variation in cell responsiveness (see Figure 3). We found that 78.7% of the total variance is explained by the first two components (PC1 = 46.2% and PC2 = 32.5%, respectively). The CNFPA distribution shows a separation between two main groups, one group is described by reduced spontaneous activity of calcium and a multi-response; in contrast, the other group is characterized by important spontaneous activity, as well as a higher potential for a response to several secretagogues. We then compared this pattern with the qualitative data associated with the patients (Table 1) and found that the two groups strongly correlated with tumor recurrence (see Figure 3E).

### 2.5. Gene Expression of Pituitary Hormone, Hypothalamic Hormone Receptors, and Signaling

We analyzed an open access scRNA-seq database, previously performed on the human pituitary gland and on several pituitary adenomas [22], to compare the expression of hormones and receptors for the hypothalamic factors evaluated in our Ca^2+^_i_ mobilization experiments.

Specifically, we analyzed the expression patterns of hormones and receptors at the single-cell level in CNFPAs and normal pituitary tissue to corroborate the existence of multi-hormonal and multi-responsive cells. From the cluster analysis, we were able to recognize well-characterized pituitary cell types in normal tissue, as they express specific hormones, except *TSHB* in thyrotrophs, probably reflecting that they are the smallest cell population in the gland (Figure 4A). We observed that hormone expression on CNFPA cells is mainly restricted to *CGA*, *LHB*, and *FSHB* subunits (Figure 4B), consistent with the observation that the *NRA51* gene, a lineage-restricted pituitary transcription factor of gonadotrophs and CNFPA, is overexpressed in CNFPA cells (see Figure 5B) [21,23].

For the receptor identification of hypothalamic secretagogues expressed by secreting-cell types, we extracted the cell types using one of the six hormone-expressed genes as a cell filter (Figure 4A,B). In normal tissues there is a tendency to express the canonical receptor of each cell type, but apparently the *GHRH* receptor is expressed in all hormone-expressing cell types (Figure 4C). Interestingly, CNFPA expresses the *GNRHR* gene in *PRL*-expressing cells, but not in cells expressing *LHB* and *FSHB*, which is the canonical co-expressing combination. A second hypothalamic hormone receptor found in CNFPAs was *TRHR* expressed in *PRL*-, *LHB*-, and *FSHB*-expressing cells. Furthermore, when we filtered the cells by their receptors, we found that *LHB* and *FSHB* cells expressed all the receptors for the hypothalamic factors (Figure 4D), except for the *CRH* receptor, which was neither expressed in normal tissue nor in CNFPAs, although *POMC* cells are present (Figure 4A). These results corroborate our finding that CNFPA cells, although monoclonal in origin, are able to express and respond to multiple hypothalamic factors.

In addition, as Ca^2+^_i_ mobilization depends on changes in the cell membrane potential due to the activity of voltage-gated Ca^2+^ and potassium channels, we explored the expression of genes that might be involved in spontaneous Ca^2+^ activity (Figure 5), as well as the cell potential to respond to multiple secretagogues. Two genes of voltage-gated Ca^2+^ (*CACNA2D1* and *CACNA2D3*) and three of potassium channels (*KCNA3*, *KCNA4*, and *KCNA5*) showed significant expression changes that may enhance the cell’s ability to depolarize (Figure 5C). Other genes overexpressed in CNFPAs were NRG1, NCAM1, GRIA2, GRM8, and EPHB6 that are known to play an important role in cell adhesion, communication, and excitability (Figure 5D). Finally, although we did not find the expression of THSB, other genes encoding proteins that could serve as thyrotroph markers are expressed in both normal tissues and CNFPAs. *GATA2*, *ISL1*, and *FOXL2* are expressed in a few cells and at low levels in both normal tissue and CNFPAs. PITX1 is expressed in CNFPAs and non-tumoral pituitary cells, but the number of cells and the level of expression is markedly higher in CNFPAs. *PITX2* is expressed almost exclusively in CNFPAs, probably reflecting the loss of cell specificity in these adenomas (Figure 5E).

## 3. Discussion

Pituitary adenomas are benign tumors that cause morbidity due to the excess of hormone secretion, hypopituitarism, and the effect of the tumor on adjacent brain areas due to mass expansion. Tumor growth depends on increased cell cycle events and the induction of angiogenesis, which creates an environment for cell proliferation. Our findings show that the relationship between CNFPA cells and the vascular system is heterogeneous among tumors, indicating that paracrine communication is also variable. Hypothalamic secretagogues, as well as some drugs, are highly dependent on the vascular system, but so is the release of hormones by the pituitary gland and CNFPAs [7,24,25,26,27]. Further investigations are needed to explore how these differences in vascular cell relation and responsiveness arise in CNFPAs and their possible correlation with agonist resistance, which can be extrapolated to other kinds of pituitary tumors, as well as to other types of endocrine and non-endocrine tumors, and even cancer.

Our results are in agreement with the previously reported data on the prevalence of CNFPA types [28,29,30]. According to Spada et al., although these tumors are highly heterogeneous in terms of secretion and morphology, they have a monoclonal origin that allows adenomas to be classified into five types [31]. CNFPAs are predominantly of gonadotroph differentiation, as they immunostain for CGA, LHβ, FSHβ, and/or NR5A1 gene products [21]. The mRNA expression of these genes has been recently corroborated by scRNA-seq analysis [22]. Other less frequent CNFPAs are null cell adenomas, which do not express any hormones or transcription factors. Silent pituitary adenomas immunostain for ACTH and TBX19 (silent corticotroph adenomas), or for PRL, or GH, and POU1F1 (silent lactotroph or somatotroph adenomas, respectively) [32,33].

Calcium regulates several cellular processes, including hormone secretion, proliferation, migration, gene expression, and apoptosis, through changes in the intracellular concentration, with a specific tempo and mode [34]. Cell signaling in the pituitary gland is stimulated by the hypothalamus, following the stimulus–secretion coupling model, where a hypothalamic factor releases a pituitary hormone [7]. In this regard, the potential of CNFPAs to express more than one receptor for hypothalamic secretagogues and respond to them, assessed in this study by the intracellular calcium activity and RNA expression at the single cell level, is not consistent with the classical stimulus–secretion coupling model, since adenomas such as gonadotropinomas are expected to respond principally, if not only, to GnRH. Of the tumors analyzed here, we found that 1 out 13 was predominantly mono-responsive (96.3%), but produced GH, PRL, LH, FSH, and ACTH, and expressed POU1F1. For the rest of the CNFPAs, we found some degree of multi-responsiveness to secretagogues, in different combinations. Additionally, cells mobilize calcium as an inherent process without stimulation from the hypothalamus and there is variation between tumors in terms of the proportion of cells with spontaneous calcium activity and the frequency of events over time.

This variation in intracellular calcium activity may further explain the clinicopathological behavior, diversity, and progression of CNFPAs. Senovilla and collaborators proposed that pituitary gland plasticity mechanisms, such as transdifferentiation, paradoxical secretion, and multi-responsive cells, which are so important during physiological challenges, are recruited during tumor development in a different manner, culminating in homeostasis disruption [11]. Furthermore, it has been suggested that changes in calcium signaling and ion channel expression may be directly associated with cell proliferation and survival, resulting in specific degrees of cell migration, resistance to apoptosis, tumor invasiveness, and recurrence [21,35]. Hence, the tumor composition of mono- and multi-responsive cells may represent different states of adenoma evolution, where the mono-response of cells indicate null or limited cell differentiation, but the tumor may evolve into a state of high cell transdifferentiation or dedifferentiation, losing phenotypic specialization and increasing its multi-responsiveness, with various clinical consequences. Importantly, a major association found in this work is the positive correlation between intracellular calcium mobilization and tumor recurrence.

Despite significant advances in the molecular biology of pituitary tumors, we still lack a reliable biomarker that could predict the clinical and biological behavior of these neoplasms [23,33,36]. Our results highlight a mechanism that could be used to gain insight into the potential recurrence of CNFPAs. For example, post-operative radiological follow-up of CNFPA patients with a high proportion of multi-responsive cells may be performed more frequently because of their potential recurrence and the absence of clinical symptoms, despite their development. Recurrent adenomas have spontaneous intracellular calcium activity, with more than 20% of multi-responsive cells and less than 50% of non-responsive cells compared to the response values of non-recurrent tumors. Unfortunately, due to technical restrictions in regard to tissue samples, we were unable to perform an experiment that may reveal the molecular identity of recurrent CNFPAs from these patients and explain the mechanistic modifications of the machinery of calcium signaling and gene expression of these adenomas. Our research group has identified some elements of the molecular alterations in other studies of pituitary adenomas, including CNFPAs [14,21,37,38]. However, further investigation is needed that traces, longitudinally, the evolution of adenoma recurrence, from the primary tumor to subsequent events of recurrence in patients, and compare them with non-recurrent CNFPAs and healthy tissues.

## 4. Materials and Methods

### 4.1. Obtention of Tissue Samples and Processing

All patients were diagnosed, treated, and received follow-up assessments at Hospital de Especialidades, Centro Médico Nacional Siglo XXI by the non-functioning adenoma clinic as part of the Endocrinology Service. The demographic, clinical, hormonal, and imaging characteristics of the patients are summarized in Table 1.

After tumor extraction, the tissue was transported in the cold organ preservation medium, CUSTODIOL^®^ (in mM: 15 NaCl, 9 KCl, 1 Alpha monopotassium ketoglutaric acid, 4 MgCl_2_ hexahydrate, 18 L-Histidine monohydrochloride monohydrate, 180 L-Histidine, 2 L-Tryptophan, 30 Mannitol, 0.015 CaCl_2_ dihydrate, 1000 mL vehicle g.s.). No more than 30 min elapsed between the surgery and the beginning of the experimental protocol. Pituitary tumors were divided into two sections with the aid of a scalpel; the first section was isolated to record the cell activity of intracellular calcium and the association of these cells with the vasculature. The second section was paraffin embedded and taken for routine characterization by immunohistochemistry using specific antibodies for the pituitary hormones (TSH, GH, PRL, FSH, LH, and ACTH) and transcription factors (NR5A1, POU1F1, and TBX19) [21,39].

### 4.2. Intracellular Calcium Activity

For the Ca2+i measurement, tissues were incubated (37 °C, 95% O_2_, and 5% CO_2_) for 30 min with the calcium sensor Fluo-4 AM (Invitrogen, Waltham, MA, USA) at a final concentration of 22 μM in 0.1% dimethylsulfoxide (DMSO) (Sigma-Aldrich, Saint Louis, MO, USA), 0.5% pluronic acid F-127 (Sigma-Aldrich, Saint Louis, MO, USA), and ACSF solution (containing in mM: 18 NaCl, 3.004 KCl, 2.501 CaCl_2_, 25.207 MgCl_2_, 2.499 NaHCO_3_, 11 glucose, 1.087 HEPES, pH 7.4). Then, the tumor was placed on top of a plexiglass chamber previously treated with 0.3% poly-L-lysine (Sigma-Aldrich, Saint Louis, MO, USA), which was then attached to the microscope stage and was continuously perfused (2 mL/min) with ACSF at room temperature.

Ca^2+^ imaging was performed as described previously [40]. The tissue was analyzed with an epifluorescence microscope (Leica M205FA; Leica Microsystem, Wetzlar, Germany), equipped with a PlanAPO 2.0× objective lens (0.35NA). Images were acquired every 200 ms with a cooled CCD camera (Coolsnap HQ; Roper Scientific Photometric, Tucson, AZ, USA). Cells loaded with Fluo-4 AM were monitored using a mercury lamp and a 488 nm filter for excitation and a 510 nm filter for emission.

The baseline activity of intracellular Ca^2+^ was recorded for 3 min, while the sample was continuously perfused with ACSF. Once the baseline activity recording was done, five independent recordings were performed with hypothalamic secretagogues (10 nM in ACSF: CRH, GHRH, GnRH, TRH, TRH-DA; BACHEM; product numbers 4011473, 4011472, 4033013, 4038214, respectively; dopamine hydrochloride is a MERK product with number H8502) applied for 30 s and leaving them for a 15 min wash between them. Secretagogues and ACSF solutions were applied by perifusion. We used a combination of 10 nM TRH and 2 μM DA to stimulate and differentiate cells expressing the DA receptor and the TRH receptor from cells only expressing the TRH receptor, representing the phenotypes of lactotrophs and thyrotrophs, respectively. Finally, to determine cell viability, a 30 s stimulus with a high potassium solution (in mM: 50 KCl, 120 NaCl, 10 HEPES, 2 CaCl_2_, pH 7.4) was applied to the tissue preparation and recorded.

To obtain numerical values on the fluorescence intensity corresponding to the Ca^2+^_i_ changes over time, the regions of interest (ROIs) were selected manually by taking the high potassium responsive cells as 100% of the analyzed population and processed using ImageJ (NIH). The values on the fluorescence intensity were then calculated using Igor Pro (Wavemetrics Inc., Portland, OR, USA), with a semi-automatic routine written by Pierre Fontanaud (Institute of Functional Genomics, Montpellier, France) for photobleach correction and to obtain normalized values on the fluorescence intensity (ΔF/Fmin), which were then plotted with a routine written by Leon Islas (Faculty of Medicine, UNAM, Ciudad de Mexico, Mexico) to identify the single cell activity over time.

### 4.3. Vascular Tracing

After incubation with Fluo-4 AM, the pituitary tissues were incubated (37 °C, 95% O_2_, and 5% CO_2_) with lectin-rhodamine (100 μg in 100 μL ASCF; Vector Laboratories, Newark, CA, USA) in order to label the vascular system and evaluate its relationship with the pituitary cells. Lectin-rhodamine was imaged, as described above, using a 510 nm filter for excitation and a 590 nm filter for emission.

### 4.4. Single Cell Analysis of RNA Expression in CNFPAs and Pituitary Cells

The scRNA-seq datasets used in this study were obtained from the Gene Expression Omnibus (GEO), with accession number GSE208108, and were previously published [22]. Gene expression analysis was performed with the filtered feature matrices of GSM6337433, GSM6337435, and GSM6337437, corresponding to the normal pituitary tissue, and GSM6337433 for the CNFPA tissue, using the Seurat R package [41]. We excluded cells, with more that 10% of mitochondrial RNA, when the number of genes was lower than 500 and when the unique molecular identifiers were lower than 800, from further analysis. Data on gene expression were normalized using the NormalizeData function in Seurat. For finding clusters in Seurat, we used the default clustering method, the original Louvain algorithm with a resolution of 1 and using 20 dimensions from the previous PCA analysis in order to calculate the UMAP. Then, clusters were curated and integrated using the FindIntegrationAnchors and IntegrateData functions to allow a comparison of the cell type identity between the samples. Cell type identification in clusters was performed manually using the Cell Marker 2.0 tool (http://bio-bigdata.hrbmu.edu.cn/CellMarker/index.html, accessed on 20 November 2023) and based on gene markers of specific pituitary cell types, previously published for both mice and human RNA expression [21,22,42].

### 4.5. Data Analysis

The response of cells to hypothalamic factors was identified by changes in the [Ca^2+^]_i_ over time and classified according to the response to one or more secretagogues as follows. Non-responding cells are those that do not respond to any secretagogue, but they are depolarized by the high potassium solution. Mono-responsive cells are those that are stimulated by one hypothalamic factor only. Multi-responsive cells are those cells that respond to two or more hypothalamic factors. Finally, the presence or absence of basal activity of Ca^2+^_i_ was identified for each cell by observing the increase in calcium activity in relation to Fmin. All statistical analysis was performed using Igor Pro (version 5) and R (The R project, version 3.5.1), along with the pheatmap, factoextra, Seurat, and ggplot2 packages.

## 5. Conclusions

It is clearly evident in the literature that Ca^2+^-permeable channels, transporters, and pumps play important roles in a wide range of tumor-related processes. The remodeling of these Ca^2+^ tool kits contribute to Ca^2+^ homeostasis dysregulation, and both regulate several of the well-known cancer hallmarks. In this work, we present strong evidence for using three parameters of the intracellular calcium response, spontaneous activity, null, and the multi-responsiveness response to secretagogues, of anterior pituitary tumor cells as a prognostic tool, as well as a guide for the treatment of CNFPAs.

## Figures and Tables

**Figure 1 ijms-25-03968-f001:**
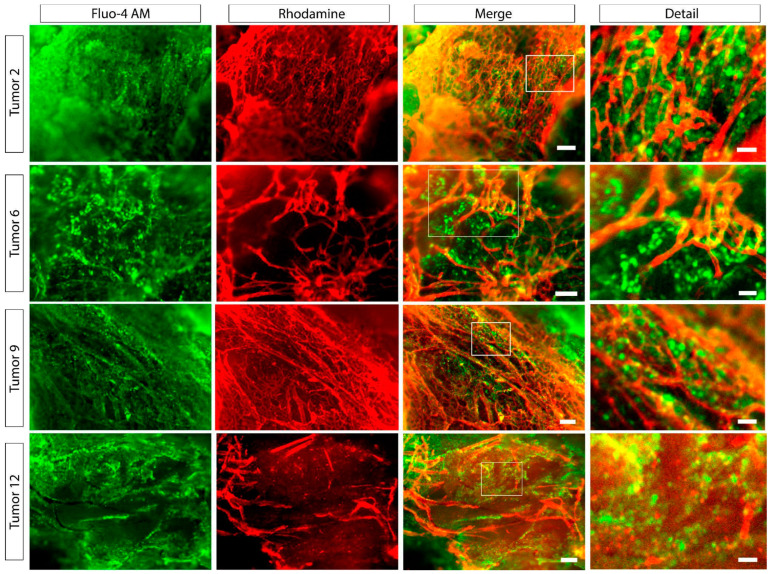
**Vascular organization in CNFPAs.** Representative images of endocrine cells (green) loaded with the calcium sensor fluo-4 AM and vascular labeling (red). The images show tumors whose vasculature is disorganized with a center from which long and thin capillaries depart (tumors 6 and 12), and vasculature with honeycomb-like organization (tumors 2 and 9). The scale bar represents 200 and 20 μm for the merge and detail columns, respectively.

**Figure 2 ijms-25-03968-f002:**
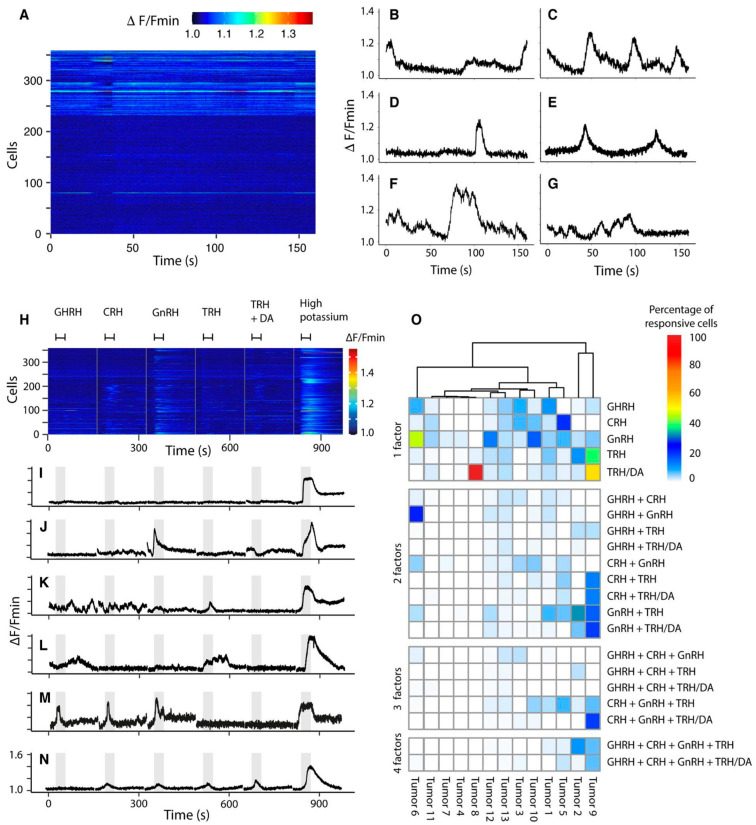
**Spontaneous and stimulated intracellular Ca^2+^ activity in CNFPAs**. (**A**) Raster graph of a tumor cell population representing the variation in Ca^2+^_i_ cell activity (360 cells). The change in fluorescence is produced by the increase or decrease in [Ca^2+^]_I_; each line on the *y*-axis represents the Ca^2+^ activity of a single cell recorded over time (*x*-axis). (**B**–**G**) Representative traces of spontaneous Ca^2+^ activity of individual cells showing variation in the number of calcium events. (**H**) Raster graph of the calcium cell activity (same cells presented in (**A**)). Increase in [Ca^2+^]_i_ is activated by hypothalamic secretagogues. Tissue samples were stimulated for 30 sec with hypothalamic factors at a concentration of 10 nM (indicated by horizontal bars above the plot). (**I**–**N**) Representative traces of intracellular Ca^2+^ activity of individual cells in response to stimuli (gray bars); (**I**) cell with no response to secretagogues but with membrane potential depolarization with a high potassium solution; (**J**) an example of mono-responsive cells with a response to GnRH but not to other stimuli; (**K**) a multi-responsive cell with a response to GHRH, CRH, TRH, and DA inhibition; (**L**) a multi-responsive cell with a response to GHRH, TRH, and DA inhibition; (**M**) a multi-responsive cell responding to GHRH, CRH, and GnRH; (**N**) a multi-responsive cell responding to CRH, GnRH, and TRH without DA inhibition. (**O**) Heatmap plot of the proportion of cells that responded to the stimuli, with hypothalamic factors for each tumor. The main factors that cause [Ca^2+^]_i_ mobilization in mono-responsive cells are TRH, GnRH, and GHRH, but multi-responsive cells are principally activated by TRH and to a lesser extent by TRH in combination with GnRH or CRH. Tumor association according to their calcium mobilization was performed using the Pheatmap package for correlation analysis. We normalized our data by taking cells that responded to high potassium as 100% in each tumor.

**Figure 3 ijms-25-03968-f003:**
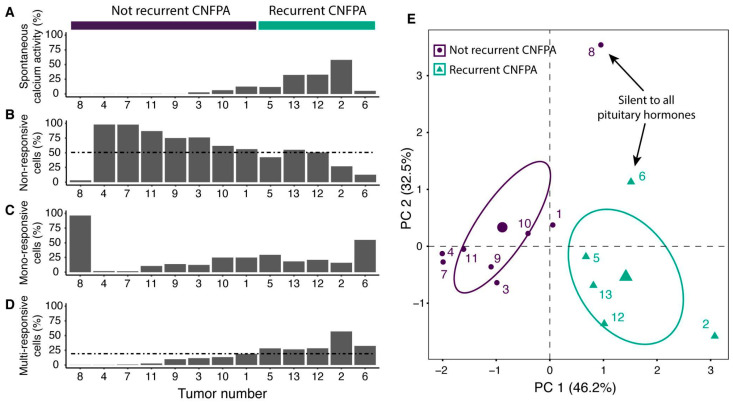
**Patterns of spontaneous and stimulated intracellular Ca^2+^ response elicited by recurrent and non-recurrent CNFPAs**. (**A**) Proportion of cells with spontaneous [Ca^2+^]_i_ activity per tumor; top color bars indicate whether tumor is not recurrent (dark blue) or recurrent (green). Tumors are grouped according to the predominant type of response of cells stimulated by hypothalamic factors: (**B**) cells not responding to any of the hypothalamic factors are non-responsive, (**C**) cells responding to only one stimulus are mono-responsive, and (**D**) cell response to two or more hypothalamic factors are considered as multi-responsive cells. More than half of the tumors analyzed (top graphs) are tumors whose largest population of cells are non-responsive (greater than 50%, dashed line). Below, we present the tumors that are mostly mono-responsive, multi-responsive, or that have similar proportions of all three types of response, except for tumor 8 that was mainly mono-responsive. (**E**) Principal component analysis reveals that CNFPA heterogeneity in the cell response is explained by the two principal components that are associated with spontaneous Ca^2+^ activity and multi-responsiveness. Variation in the response is also correlated with the recurrence of adenomas. Principal component analysis was performed using the factoextra package. Data has been normalized to the cells that responded to high potassium in each tumor.

**Figure 4 ijms-25-03968-f004:**
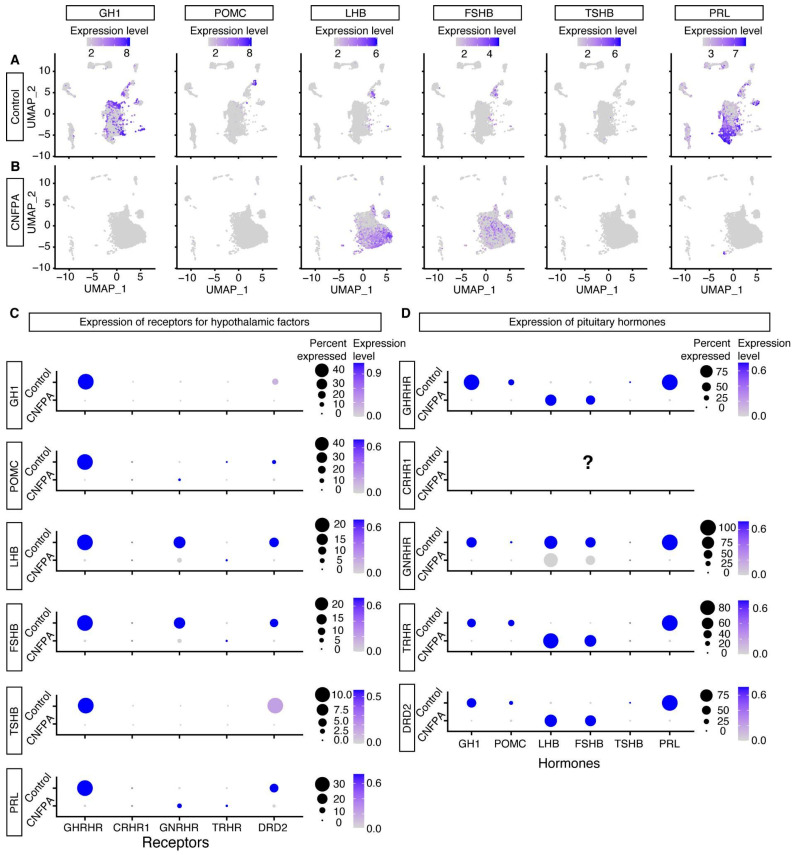
**Expression of multiple receptors for hypothalamic hormones in pituitary cell types and multi-hormonal cells.** (**A**,**B**) Uniform manifold approximation and projection (UMAP) plot of pituitary transcriptome for normal tissue and CNFPAs showing gene expression of pituitary hormones. (**C**) Dot plot showing the proportion and expression quantity of receptors to hypothalamic factors, according to hormone-specific cell types. Cells were filtered according to the expression of one of the pituitary hormones. Cells from control tissue express the canonical receptors for each hormone-expressing cell type, with few cells expressing different receptors, with the exception of the GHRH receptor. CNFPA expresses mainly GnRHR and TRHR. (**D**) When cells were filtered by the expression of one of the receptors for hypothalamic hormones, we observed diversity in the hormone contents, indicating that one cell expressing a particular hormone may express several receptors. We found no cells expressing the CRHR1 gene, which is known to be expressed by corticotrophs.

**Figure 5 ijms-25-03968-f005:**
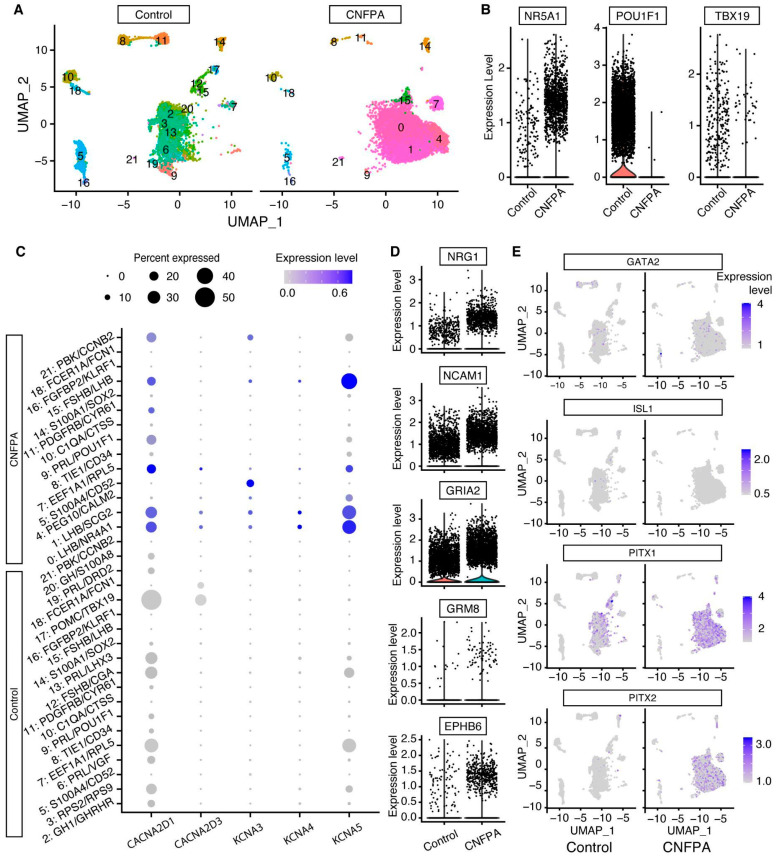
**Differential expression of genes involved in cell excitability in normal pituitary cells and CNFPAs.** (**A**) UMAP plot of normal tissue and CNFPA cell transcriptomes; colors indicate cell clusters that were identified by markers known to be expressed in the pituitary cell types listed in the *y*-axis of the dot plot in (**C**). (**B**) The expression of the three lineage markers of secretory cell types in pituitary cell NR5A1 is highly expressed in CNFPAs. (**C**) Differential expression of selected voltage-gated potassium and calcium channels that may be correlated with multi-responsiveness and the generation of spontaneous calcium events in CNFPAs. (**D**) Violin plots showing the expression of genes involved in cell communication and the regulation of membrane potential. (**E**) Distribution of gene markers that are known to be expressed in thyrotrophs.

**Table 1 ijms-25-03968-t001:** Main characteristics of the patients and tumors analyzed in this study.

Tumor Number	Age/Gender	Maximum Diameter (cm)	IHC Phenotype	Hormone Deficiency	Recurrent?/Number of Events ^1^	Type of Surgery and Post-Treatment ^2^
1	38/F	2.3	Silent GH adenoma POU1F1 positive	TSH	No	TSS
2	65/M	3	Gonadotropinoma NR5A1 positive	TSH, ACTH	Yes/6	TSS, XRT, CBG, TMZ
3	52/F	6.2	Silent PRL/ACTH adenoma TBX19 positive	TSH, ACTH	No	TSS, XRT, CBG
4	49/F	2.7	Gonadotropinoma NR5A1 positive	TSH	No	TSS
5	63/M	5.4	Gonadotropinoma NR5A1 positive	All	Yes/2	TSS, TC
6	58/F	4.4	Silent GH/PRL/LH/FSH/TSH/ACTH adenoma NR5A1 positive	TSH	Yes/2	TSS
7	73/F	3.3	Silent GH/PRL/LH/FSH adenoma NR5A1 positive	TSH, ACTH	No	TSS, XRT
8	66/F	2.6	Silent GH/PRL/LH/FSH/ACTH adenoma NR5A1 positive	None	No	TSS
9	60/M	5.6	Gonadotropinoma NR5A1 positive	All	No	TC
10	47/F	3.9	Gonadotropinoma NR5A1 positive	TSH	No	TSS
11	60/M	3.8	Gonadotropinoma NR5A1 positive	All	No	TSS
12	47/M	7	Gonadotropinoma NR5A1 positive	All	Yes/1	TC, CBG
13	45/F	3.5	Silent ACTH adenoma TBX19 positive	TSH	Yes/1	TSS

^1^ The number of recurrent events is indicated, excluding the first surgery of patients, which is considered as a primary tumor. For each patient, the last event of recurrence is the CNFPA evaluated and analyzed in this work. ^2^ Treatments for patients were as follows: transsphenoidal surgery (TSS); transcranial surgery (TC); postoperative radiation therapy (XRT); cabergoline (CBG); temozolomide (TMZ).

## Data Availability

All data generated are presented in this work and raw data are available upon request.

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
