# Peer review of "Association between Intracellular Calcium Signaling and Tumor Recurrence in Human Non-Functioning Pituitary Adenomas"

_ijms, 2024, doi:10.3390/ijms25073968_

Round 1

Reviewer 1 Report

Comments and Suggestions for Authors

1, Authors need to conduct some experimental validation, such as real-time PCR, for expression analysis as depicted in Figure 5

2.The abstract needs to be updated by the authors in light of the specific findings. The present version of the abstract makes it look as if I am reading the abstract of a review article.

3. Figure quality needs to be improved, and also the font size of figure legends in all figures should be increased

Comments on the Quality of English Language

NA

Author Response

March 11, 2024

Dear reviewers,

Thank you very much for your valuable comments on the manuscript entitled Association between intracellular calcium signaling and tumor recurrence in human non-functioning pituitary adenomas. We have modified the manuscript according to your suggestions as follows.

Reviewer 1

Comment 1. Authors need to conduct some experimental validation, such as real-time PCR, for expression analysis as depicted in Figure 5.

Answer. At the time of surgical procedures we obtained a section of the CNFPA that was used for the calcium recording experiment. And the other section of tissue was handled by technicians of the hospital in order to determine the phenotype of the tumor and other clinical analysis with medical purposes. For that reason we were unable to perform other experiments on the samples. On the other hand, our group has published some articles analyzing the identity of molecular activity and gene expression using several pituitary adenomas, including CNFPA, and are now cited and emphasized in the discussion and added to bibliography. However, those analyses do not correspond to the samples we used for calcium imaging and can not directly correlate to our result or even be edited as they have already been published. But we strongly believe, and we noted that in our discussion, that it may be important in the future to evaluate the evolution of CNFPA, taking advantage of the available samples of recurrent adenomas from patients that have a long medical history at the hospital. In this way, we may be able to understand the molecular modifications through the years from a single patient and compare them with no recurrent adenomas and healthy samples. We also highlighted these comments at the end of the conclusions of the manuscript.

Comment 2.The abstract needs to be updated by the authors in light of the specific findings. The present version of the abstract makes it look as if I am reading the abstract of a review article.

Answer. We have added emphasis in the abstract to remark that this is an original article and also its respective results and conclusions accordingly. We excluded the idea of writing headings in the abstract for the main sections of the article as the IJMS invites the authors to not do that. Changes were highlighted in the abstract section of our manuscript.

Comment 3. Figure quality needs to be improved, and also the font size of figure legends in all figures should be increased.

Answer. Our original images possess a high resolution quality (300 dpi) and were uploaded independently during the submission process in a zip file. The .doc file of the manuscript was submitted with JPEG images and may be altered in resolution after submission. We have updated the images and increased the font size to a minimum of 8 pt in all figure legends in order to increase the clarity of information. These figures were updated in the IJMS platform and may, for a faster analysis, be found in the following links as well:

  • Figure 1: https://drive.google.com/file/d/1d9MoFaSj7UXh8hPScJFwa7tfcfIHzEpt/view?usp=sharing
  • Figure 2: https://drive.google.com/file/d/1HMZ8-szqnFznXcCMfgh6_3PC-8ZkHJtl/view?usp=sharing
  • Figure 3: https://drive.google.com/file/d/1b3eEYMD-uWTafFwos1d9l7A0iQOhk2pW/view?usp=sharing
  • Figure 4: https://drive.google.com/file/d/1sv3Ql_GAIuDIT1u7ZzUaAB4KroJwXm3V/view?usp=sharing
  • Figure 5: https://drive.google.com/file/d/1eUMx0wkRJfu6L9t8jNhnb17vf3s-uVMa/view?usp=sharing

Reviewer 2 Report

Comments and Suggestions for Authors

This research article reported a promising approach that could identify the recurrence phenotype of nonfunctioning pituitary adenomas (CNFPA). It is confused through the evaluation method of tumor receptor expression patterns in characterizing the recurrence of CNFPA. Better than IHC or molecular (gene/protein) evidence, Calcium imaging may help to identify the recurrence from the pathophysiology way. I agree with this idea in the physiological aspect, however, one major question on my side should be considered and highlighted in the result and also the discussion section. How do the post-treated conditions in patients contribute to Calcium signals in CNFPA? Other minor comments may need addressing to qualify the manuscript for publication.

Minor comments,

1.     Page 8, the authors did not show Fig 3H. “groups strongly correlated with tumor recurrence (See figure 3H).

2.     Figures have low resolution and cannot show clear information.

Author Response

March 11, 2024

Dear reviewers,

Thank you very much for your valuable comments on the manuscript entitled Association between intracellular calcium signaling and tumor recurrence in human non-functioning pituitary adenomas. We have modified the manuscript according to your suggestions as follows.

Comment 3. This research article reported a promising approach that could identify the recurrence phenotype of nonfunctioning pituitary adenomas (CNFPA). It is confused through the evaluation method of tumor receptor expression patterns in characterizing the recurrence of CNFPA. Better than IHC or molecular (gene/protein) evidence, Calcium imaging may help to identify the recurrence from the pathophysiology way. I agree with this idea in the physiological aspect, however, one major question on my side should be considered and highlighted in the result and also the discussion section. How do the post-treated conditions in patients contribute to Calcium signals in CNFPA?

Answer. We do not have results of the effect of treatment to patients as it was applied after surgery and the analysis of calcium activity. In the case of recurrent CNFPA, we have emphasized in results that we worked specifically with the last event of recurrence. We also modified the text to make it clear that the treatment was postoperative. Modifications were highlighted in the manuscript for easy identification.
